

# Evolutionary history of the vertebrate Piwi gene family

Javier Gutierrez[1], Roy Platt[2], Juan C. Opazo[3,4,5], David A. Ray[6], Federico Hoffmann[7,8] and Michael Vandewege[1]

[1] Department of Biology, Eastern New Mexico University, Portales, NM, United States of America
[2] Host Pathogen Interaction Program, Texas Biomedical Research Institute, San Antonio, TX, United States of America
[3] Instituto de Ciencias Ambientales y Evolutivas, Universidad Austral de Chile, Valdivia, Chile
[4] Millennium Nucleus of Ion Channel-Associated Diseases (MiNICAD), Valdivia, Chile
[5] Integrative Biology Group, Universidad Austral de Chile, Valdivia, Chile
[6] Department of Biological Sciences, Texas Tech University, Lubbock, TX, United States of America
[7] Department of Biochemistry, Molecular Biology, Entomology, and Plant Pathology, Mississippi State University, Starkville, MS, United States of America
[8] Institute for Genomics, Biocomputing and Biotechnology, Mississippi State University, Starkville, MS, United States of America

## ABSTRACT

PIWIs are regulatory proteins that belong to the Argonaute family. Piwis are primarily expressed in gonads and protect the germline against the mobilization and propagation of transposable elements (TEs) through transcriptional gene silencing. Vertebrate genomes encode up to four Piwi genes: Piwil1, Piwil2, Piwil3 and Piwil4, but their duplication history is unresolved. We leveraged phylogenetics, synteny and expression analyses to address this void. Our phylogenetic analysis suggests Piwil1 and Piwil2 were retained in all vertebrate members. Piwil4 was the result of Piwil1 duplication in the ancestor of gnathostomes, but was independently lost in ray-finned fishes and birds. Further, Piwil3 was derived from a tandem Piwil1 duplication in the common ancestor of marsupial and placental mammals, but was secondarily lost in Atlantogenata (Xenarthra and Afrotheria) and some rodents. The evolutionary rate of Piwil3 is considerably faster than any Piwi among all lineages, but an explanation is lacking. Our expression analyses suggest Piwi expression has mostly been constrained to gonads throughout vertebrate evolution. Vertebrate evolution is marked by two early rounds of whole genome duplication and many multigene families are linked to these events. However, our analyses suggest Piwi expansion was independent of whole genome duplications.

# INTRODUCTION

The Piwi gene family has become one of the more charismatic gene families in recent decades due to their role in defending the genome against transposable elements (TEs) and viruses (*Siomi et al., 2011*; *Sun et al., 2017*; *Kolliopoulou et al., 2019a*; *Kolliopoulou et al., 2019b*). PIWI proteins belong to a larger family of Argonaute proteins that include two major

Corresponding author
Michael Vandewege,
mike.vandewege@gmail.com

clades, the AGOs and PIWIs (*Carmell et al., 2002*). All members of the Argonaute family contain a PAZ and PIWI domain and participate in RNA-induced silencing complexes (RISCs) with small RNAs. AGOs are ubiquitously expressed and bind microRNAs (*Tang, 2005*) to target and silence mRNAs. In vertebrates, PIWI proteins are generally restricted to the germline, bind their own class of small RNAs, PIWI interacting RNAs (piRNAs), and regulate TEs and viruses through transcriptional silencing (*Girard et al., 2006*; *Lau et al., 2006*; *Brennecke et al., 2007*; *Sun et al., 2017*). The details of PIWI function vary among animals, but most seem to be involved in a model described as the 'ping-pong cycle' where PIWIs cleave RNA into piRNAs, the cleaved piRNAs form riboprotein complexes with the PIWIs and direct these complexes to bind complementary transcripts. These complexes can then cleave newly bound transcripts and repeat the cycle. Secondary piRNAs can also be used to guide PIWI proteins to TE loci in the genome and initiate methylation (*Aravin et al., 2008*; *Kuramochi-Miyagawa et al., 2008*; *Kuramochi-Miyagawa et al., 2008*; *Rozhkov, Hammell & Hannon, 2013*). In addition to the ping-pong cycle, the diversity of piRNAs is increased by 5′-to-3′ phasing from the site of initial piRNA formation (*Han et al., 2015*; *Mohn, Handler & Brennecke, 2015*; *Ozata et al., 2019*).

The bulk of vertebrate PIWI protein functional analyses are restricted to experiments in mice. From these experiments, we understand that the expression of Piwis varies temporally during germ cell development and spermatogenesis. Piwil2 (Mili) is the first Piwi to become expressed at embryonic day 12.5 (E12.5) in developing testes and is linked to the post-transcriptional silencing of TEs (*Aravin et al., 2007*). Piwil4 (Miwi2) is expressed between E14.5 and post-natal day 3 (P3). piRNAs from PIWIL2 are loaded onto PIWIL4 and PIWIL4 initiates the *de novo* establishment of methylation marks among TE loci in gonocytes during this time (*Carmell et al., 2007*; *Aravin et al., 2008*; *Molaro et al., 2014*; *Zoch et al., 2020*). PIWIL4 is also expressed in undifferentiated spermatogonia in adults, although the link to TEs in these cell types is still under investigation (*Carrieri et al., 2017*; *Vasiliauskaite et al., 2018*). Piwil1 (Miwi) is the last Piwi gene expressed at approximately P14, during the pachytene stage of prophase I (*Girard et al., 2006*). However, most piRNAs associated with PIWIL1 are derived from non-coding regions and PIWIL1 plays a relevant role cleaving/removing mRNAs during later stages of spermatogenesis (*Reuter et al., 2011*; *Li et al., 2013*; *Gou et al., 2015*; *Wu et al., 2020*). Some placental mammals, including Primates and Laurasiatherians have a 4th paralog, Piwil3. Piwil3 is the least studied of the four paralogs, likely because this paralog is absent from the mouse genome. However, experiments in cattle suggest Piwil3 is largely expressed in oocytes and early embryos and PIWIL3 generates TE derived ping-pong piRNAs (*Roovers et al., 2015*; *Tan et al., 2020*).

The copy number of the Piwi gene family varies among animals, for example *C.elegans* encodes two Piwis from a lineage specific duplication (*Wynant, Santos & Vanden Broeck, 2017*), *Drosophila melanogaster* encodes three Piwis (*Lewis et al. 2018*), but some turbellaria flatworms could have up to eight lineage specific paralogs (*Fontenla, Rinaldi & Tort, 2021*). The varying rates of sequence evolution and gene turnover has made the animal Piwi phylogeny difficult to resolve and multiple contradicting Piwi gene trees that do not mirror species relationships have been presented (*Kerner et al., 2011*; *Wynant, Santos & Vanden Broeck, 2017*; *Fontenla, Rinaldi & Tort, 2021*). However, a general theme of Piwi
phylogenetics suggests there are two major subfamilies, Piwil1 and Piwil2. Some trees suggested vertebrate piwis are not directly orthologous to insect Piwis (*Wynant, Santos & Vanden Broeck, 2017*), others found that *Drosophila* Ago3 is orthologous to vertebrate Piwil2 but the Piwi and Aub genes of *Drosophila* do not have a vertebrate ortholog (*Kerner et al., 2011*; *Jehn et al., 2018*; *Fontenla, Rinaldi & Tort, 2021*).

Consistent with the two subfamily generalization, vertebrate genomes encode at least two copies of the Piwi family, a copy from the Piwil1 (-iwi) group and another from the Piwil2 (-ili) group, however vertebrates have a maximum of four Piwi paralogs (See above) and questions remain about the timing and mechanism of vertebrate Piwi duplication. Here, we are interested in leveraging phylogenetic and synteny analyses to clarify the duplication history of the vertebrate Piwi family as well as understanding the patterns of selection and expression among major lineages.

## METHODS

### Sequence acquisition

We used Ensembl v.101 (*Yates et al., 2020*) and NCBI release 239 (*Sharma et al., 2018*) to collect known Piwi and Piwi-like coding sequences (CDS) from representative species of all major groups of vertebrates. Specifically, our sampling included cyclostomes (jawless fishes; lamprey) and gnathostomes (jawed vertebrates) from chondrichthyes (cartilaginous fishes), ray-finned fishes, lobe-finned fishes and tetrapods (amphibians, reptiles, birds and mammals). To help understand the ancestral state and focus on changes that occurred within vertebrates, we included Piwis from closely related deuterostomes as outgroups (Table S1, Fig. S1). The longest CDS was selected if there were multiple transcripts for a Piwi gene. Unidentified CDSs that shared similarity to Piwis were queried against the NCBI non redundant protein database *via* BLAST (*Altschul et al., 1990*) to confirm homology. If a Piwi paralog was absent or poorly annotated in Ensembl, we queried the NCBI for the paralog using human Piwi sequences as BLAST queries.

### Phylogenetic reconstruction

All Piwi sequences were translated to amino acids and aligned using the LINSI strategy in MAFFT v7 (*Katoh & Standley, 2013*). We constructed phylogenetic trees from the amino acid alignment with two methods. First, we used IQ-Tree2 (*Minh et al., 2020*) and let IQ-Tree2 find the best fitting substitution model (*Kalyaanamoorthy et al., 2017*). Ten independent phylogenetic analyses were run simultaneously to explore tree space and we selected the tree with the highest likelihood score. Node support for the best scoring tree was evaluated with the ultrafast bootstrap method (*Hoang et al., 2018*) and the SH-like approximate likelihood ratio test (SH-aLRT) (*Guindon et al., 2010*). In both cases, we used 1,000 pseudoreplicates. Second, we used ExaBayes v1.5.1 (*Aberer, Kobert & Stamatakis, 2014*) to conduct Bayesian analyses. We ran four simultaneous chains for 1,000,000 generations sampling every 500 generations, using the same substitution model chosen by IQ-Tree2 but without additional model parameters. Chains were considered as having converged once the average standard deviation of split frequency among runs was less than 0.01. We summarized results with a majority-rule consensus tree from the set of sampled

trees after the first 25% were discarded. We repeated these tree construction analyses on a reduced alignment generated after removing poorly aligned regions using Gblocks and allowed up to five contiguous non-conserved positions and columns to contain gaps; -b3 = 5 -b5 = a (*Castresana, 2000*) to test the impact of alignment quality on our results.

## Synteny analyses

To understand the conservation of synteny as well as gene gain and loss, we examined the position of protein coding genes up and downstream of Piwi genes in representative vertebrates. We used gene coordinate information from Ensembl v.101 as well as estimates of orthology and paralogy from the Compara database (*Herrero et al. 2016*). If a Piwi gene was presumed missing, we used BLAST in Ensembl with human Piwi queries to locate any Piwi derived fragments near expected positions. If no fragments were found, we concluded the gene was missing, but if fragments were found and a full annotation was lacking, we presumed the Piwi to be pseudogenized.

## Estimating $d_N/d_S$ rates and selection tests

For each Piwi paralog in each major lineage with multiple samples (birds, mammals, and ray-finned fishes), we made independent codon alignments by translating DNA sequences to amino acid sequences, generating LINSI amino acid alignments and 'reverse translating' alignments into codon alignments using a custom Python script (available at github.com/mike2vandy/PiwiGeneFamily along with other scripts used). We specifically conducted site-tests for diversifying selection (*Goldman & Yang, 1994*) among alignments using the codeml module of PAML v4.9. We calculated the likelihood of models that allow $d_N/d_S$ to vary among codon positions (M1a, M2a, M8a and M8). We used the likelihood ratio test (LRT) to test for significant differences between nested models that do not allow selection to those that do (M1a *vs.* M2a, M8a *vs.* M8). We also estimated a single $d_N/d_S$ rate for each alignment using codeml's M0 one-ratio model and pairwise $d_N/d_S$ distances using the modified Nei-Gojobori (proportional) method in MEGA-X (*Kumar et al., 2018*). We noticed an intronic sequence that included stop codons was retained in the manatee Piwil4 sequence. We suspect this was an annotation artifact and because of it, this sequence was not included in selection tests and $d_N/d_S$ estimates.

To view changes in $d_N/d_S$ along the mammalian Piwil1 and Piwil3 branches we constructed a phylogenetic tree of mammalian Piwil1 and Piwil3 and used a free-ratio model to estimate $d_N/d_S$ rates on all branches (*Yang, 1998*). We also used the adaptive branch-site random effects likelihood (aBSREL; (*Smith et al., 2015*)) approach implemented in Datamonkey v2.2 (*Weaver et al., 2018*) which detects episodes of positive selection on all designated foreground branches. We performed a LRT between the null model ($d_N/d_S = 1$) against the alternative where a branch was undergoing some form of selection ($d_N/d_S \neq 1$). All Piwil3 branches were labeled as foreground and tested.

## Expression analyses

To assess historical changes in Piwi expression through vertebrate evolution, we collected RNASeq data from representative tissues (brain, heart, kidney, liver, ovary and testes) and species (elephant shark, spotted gar, clawed frog, spiny toad, chicken, opossum, cow,

and human) from NCBI's short read archive (SRA; *Leinonen, Sugawara & Shumway, 2011*). Accession numbers for species and tissues can be found in Table S2. For all species, we used appropriate Ensembl's CDSs downloaded from BioMart as a mapping reference. We only used the longest CDS per gene. Prior to mapping, RNASeq reads were cleaned with Trimmomatic v0.38 (*Bolger, Lohse & Usadel, 2014*) using the trimming and adapter removal parameters: HEADCROP:5, SLIDINGWINDOW:5:30, MINLEN:50, and ILLUMINACLIP:2:30:10. RNASeq reads were mapped to reference CDS libraries using default parameters of RSEM v1.3.1 (*Li & Dewey, 2011*) and Bowtie v1.2.2 (*Langmead et al., 2009*) to estimate gene expression in units of transcripts per million (TPM).

# RESULTS

## Phylogenetic analysis

Our first goal was to reconstruct the Piwi phylogeny from vertebrate representatives and deuterostome outgroups using both maximum likelihood (ML) and Bayesian strategies. We collected 168 sequences from 63 species (Table S1). We generated trees in three different strategies: a tree of select representative species (Fig. 1), a tree from all collected sequences (Fig. S2), and because the N-terminus was difficult to align, we also estimated a tree from the most alignable regions of the amino acid alignment which included the PAZ and PIWI domains (Fig. S3). All trees were relatively similar and sequences fell into the two major clades, the Piwil1 group and Piwil2 group (Fig. 1). There have been no additional duplications of Piwil2 among deuterostomes and the subfamily is monophyletic. By contrast, Piwil4 and the mammalian specific Piwil3 were derived from independent duplications of a vertebrate Piwil1 (Fig. 1). These additional duplications within the Piwil1 clade indicate that the gnathostome Piwil1 is not a 1:1 ortholog of the ancestral Piwil1 gene. However, we have retained the traditional nomenclature for simplicity.

The Piwil4 gene of vertebrates was likely derived from an ancient duplication of Piwil1 in the ancestor of gnathostomes, however a lamprey Piwil1-like sequence was found sister to gnathostome Piwil4 in both ML and Bayesian trees. Therefore, we can interpret the Piwil4 duplication in two different ways. The Piwil4 duplication occurred in the ancestor of vertebrates, but Piwil1 was lost in lamprey and the lamprey sequence is an ortholog of Piwil4. Alternatively, the Piwil4 duplication occurred in gnathostomes, but the lamprey Piwil1-like gene is artificially sister to Piwil4. Piwil4 was present in cartilaginous and lobe-finned fishes, mammals and reptiles, but was absent in ray-finned fishes and birds (Fig. 1; Fig. S1).

Our analyses suggest Piwil3 is the result of a duplication of Piwil1 in the common ancestor of therian mammals given its presence in marsupial and placental mammals and its absence from Monotreme genomes (Fig. 1). However, not all placental mammals have retained Piwil3. Piwil3 was present in marsupials (opossum and koala), Laurasiatherians (bat, dog, and cow) and most Euarchontoglires (primates, rodents and relatives), but we were unable to find it in Afrotherians (elephant and manatee), Xenarthrans (armadillo) and mouse-like rodents which included the kangaroo rat, deer mouse and house mouse (Fig. S1).

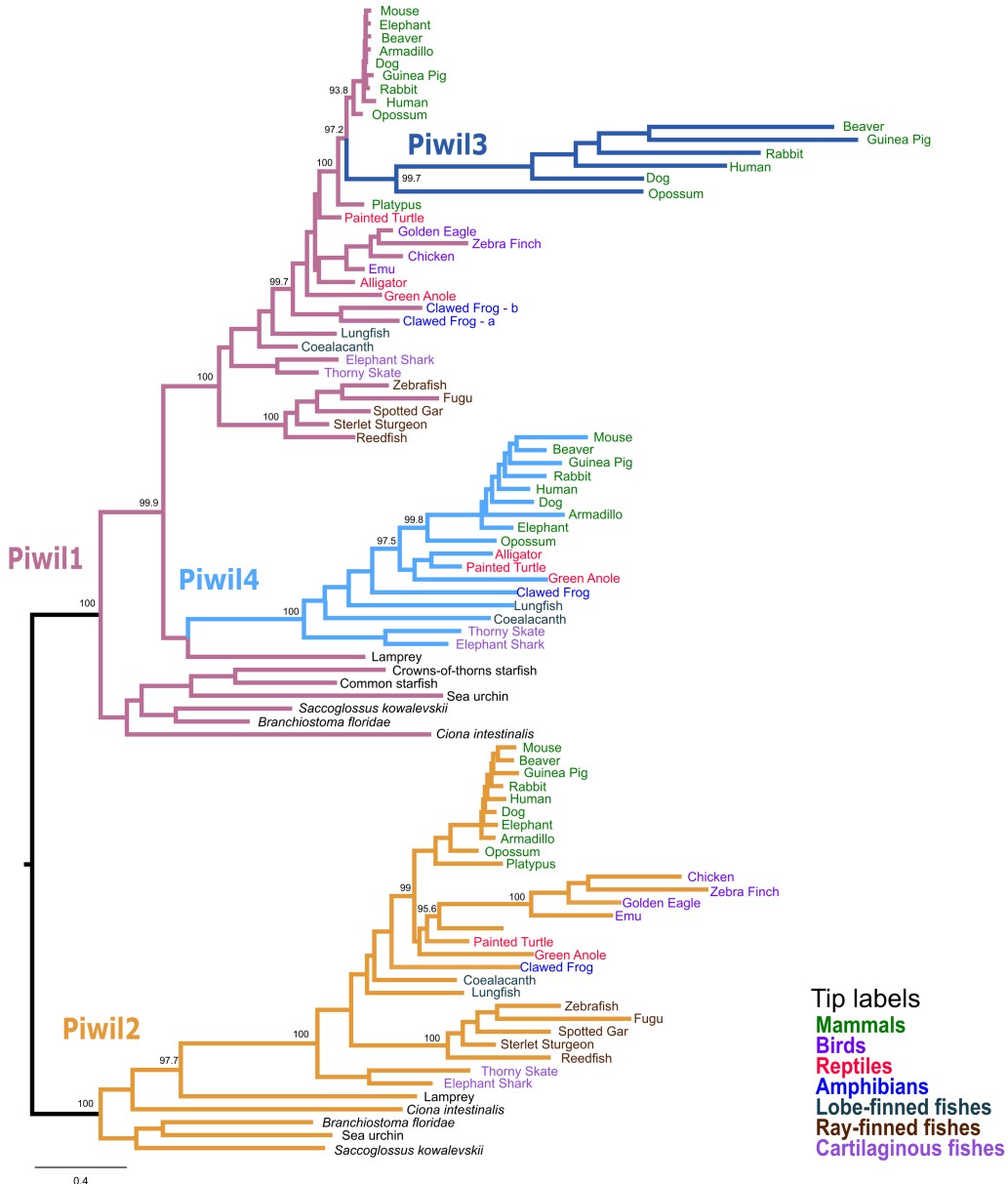

**Figure 1** **Phylogenetic reconstruction of the vertebrate Piwi family.** A phylogenetic tree constructed from a reduced dataset to summarize relationships among Piwi paralogs. The IQ-Tree2 model chosen was LG+F+I+G4 (LG model using empirical base Frequencies, a proportion of Invariable sites, and a discrete Gamma model with four rate categories). A tree constructed from all species used is presented in Fig. S2. Piwi paralogs are color coded on the tree and tip labels are color coded for major gnathostome groups. The displayed tree was the maximum likelihood tree constructed with IQ-Tree2. Numbers by nodes correspond to support values from the ultrafast bootstrap routine in IQ-Tree2.

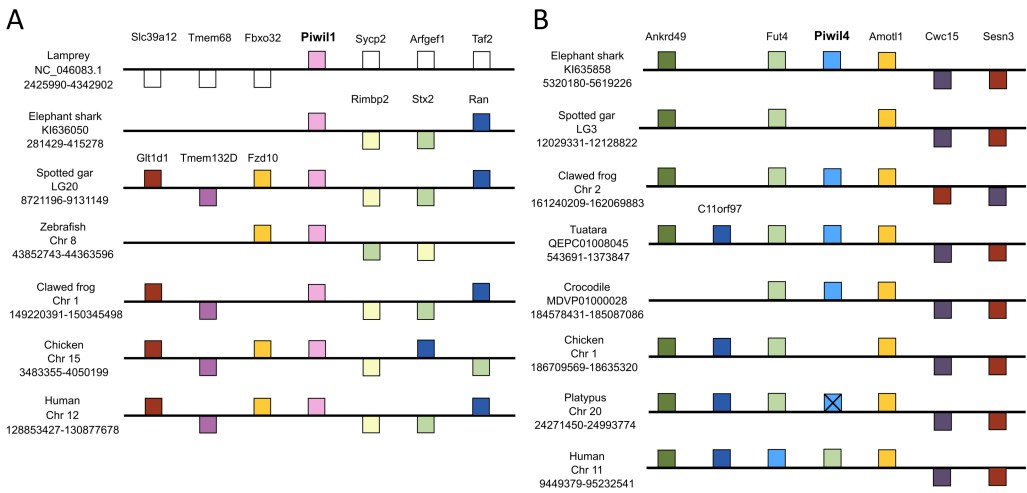

**Figure 2** **Synteny comparisons of vertebrate Piwil1 and Piwil4.** Organization of genes up and downstream of (A) Piwil1 and (B) Piwil4. Distances are not drawn to scale. White boxes represent genes that are not homologous to any other genes in the synteny block. Boxes on top of the black line reflect genes in forward orientation relative to Piwi genes and boxes below the line are in the opposite orientation. An "X" indicates pseudogenization and empty space between Fut4 and Amotl1 in 2B indicates an absent Piwil4.

An additional copy of Piwil1 (Piwil1a and Piwil1b) had been previously described in the clawed frog (*Wilczynska et al., 2009*), but our analyses also revealed this additional paralog was present in the spiny toad (Fig. S2). Interestingly, a third intact Piwil1 copy was identified in the spiny toad (Fig. S2). This copy is most likely a retrogene, diagnosed by a lack of introns in the coding sequence (Fig. S4A).

## Synteny analyses

We used genomic coordinates to identify genes up- and down-stream of Piwi genes to resolve or confirm homology and infer the duplication mechanism behind the vertebrate Piwi expansion. Synteny analyses included representative species with contiguous regions for at least 10 genes up and downstream of each Piwi. In most cases, this included the elephant shark, spotted gar, zebra fish, clawed frog and chicken and human, but varied among Piwis. The lamprey assembly on Ensembl lacked contiguity around the Piwi genes, therefore we queried sea lamprey coordinate data from a GFF file at https://genomes.stowers.org/sealamprey (*Smith et al. 2018*). Unfortunately, there was a lack of synteny among Piwi genes between cyclostomes and gnathostomes, but synteny around Piwil1, Piwil2 and Piwil4 has generally been conserved among gnathostomes (Fig. 2; Fig. S5). Synteny around Piwil2 was less conserved than Piwil1 and observed three different sets of upstream genes, however Slc39a14 and Ppp3cc were consistently downstream of Piwil2 in all lineages (Fig. S5).

Piwil1 was flanked by Rimbp2, Stx2, and Ran on the 3′end in all gnathostomes, but synteny was only conserved upstream of Piwil1 among bony and jawed vertebrates where Piwil1 is generally flanked by Fzd10, Tmem132d and Glt1d1 (Fig. 2A). By contrast, synteny around Piwil4 has been conserved throughout all gnathostome evolution. Although Piwil4
was lost in ray-finned fish and birds, the overall locus has remained intact (Fig. 3B). In the spotted gar, the distance between Fut4 and Amotl1 was 18,372 bases and there were no BLAST hits to Piwi genes in that region. The same was true for the chicken, where the distance between Fut4 and Amotl1 was 45,906 and lacked sequences similar to Piwil4. By contrast, the same region in the crocodile covers 157,135 bases and includes a Piwil4 gene (Fig. 3B). Interestingly, Piwil4 is intact in all mammals except the platypus. The Piwil4 locus in platypus does contain a lncRNA annotation (ENSOANT00000069592) between Fut4 and Amotl1 with similarity to Piwil4. However, a complete gene with an open reading frame could not be recovered. Therefore we suspect that Piwil4 has been pseudogenized in the platypus (Fig. 2B) but has not been fully purged from the genome.

Piwil1 and Piwil4 were not flanked by any co-duplicating gene families (Fig. 2), an indicative signal of whole genome or segmental duplications (*Campanini et al., 2015*; *Opazo et al., 2015*). Furthermore, both teleosts and salmonid fishes, which have experienced additional whole genome duplication events (*Jaillon et al., 2004*; *Macqueen & Johnston, 2014*; *Lien et al., 2016*), lacked any additional Piwi paralogs (Fig. S2).

Piwil3 is a direct neighbor of Piwil1 in marsupials which would suggest that the Piwil1 and Piwil3 genes of therian mammals are co-orthologs of the ancestral mammalian Piwil1 gene derived from a tandem duplication of Piwil1 in their common ancestor (Fig. 3A). Interestingly, the position of Piwil3 is conserved among marsupials, but Piwil3 is in a novel location in placental mammals where flanking genes vary among lineages (Fig. 3A). Taking these results together, the expansion of the Piwi gene family in vertebrates can best be explained by tandem duplication followed by a translocation due to lineage-specific genomic rearrangements.

## Piwil3 and selection tests

We observed very long branches among Piwil3 paralogs (Fig. 1) and decided to test the role diversifying selection has played on the evolution of Piwil3. Using PAML's free ratio model, we found that $d_N/d_S$ rates among Piwil3 sequences were considerably faster than rates among therian Piwil1 sequences (Fig. 3B). Since the free-ratio model does not offer any additional information besides changes in $d_N/d_S$ and is not statistically robust, we used site-tests among Piwil3 sequences and found that models allowing selection (M2a and M8) were significantly better fits to the data than models that disallow positive selection (M1a and M8a) (Table S3). Codons throughout the entire gene, including the conserved PAZ and PIWI domains (Fig. S6), exhibited signatures of positive selection (Fig. 3C; Fig. S7) according to Bayes Empirical Bayes (BEB) estimation of site posterior probabilities. A separate branch-site test, aBSREL in HyPhy, suggested 17 out of 22 tested Piwil3 branches have undergone episodic positive selection (Fig. S8).

We then compared Piwil3 patterns of evolution to the other paralogs within tetrapod groups (birds, mammals and ray-finned fishes). Site-tests of diversifying selection suggested no other Piwi paralog is evolving in a diversifying pattern. In all cases except Piwil3, models allowing positive selection were not improved over models restricting selection (Table S3). Further, one-ratio (Fig. 3D) and pairwise (Fig. 3E) $d_N/d_S$, although likely inflated due

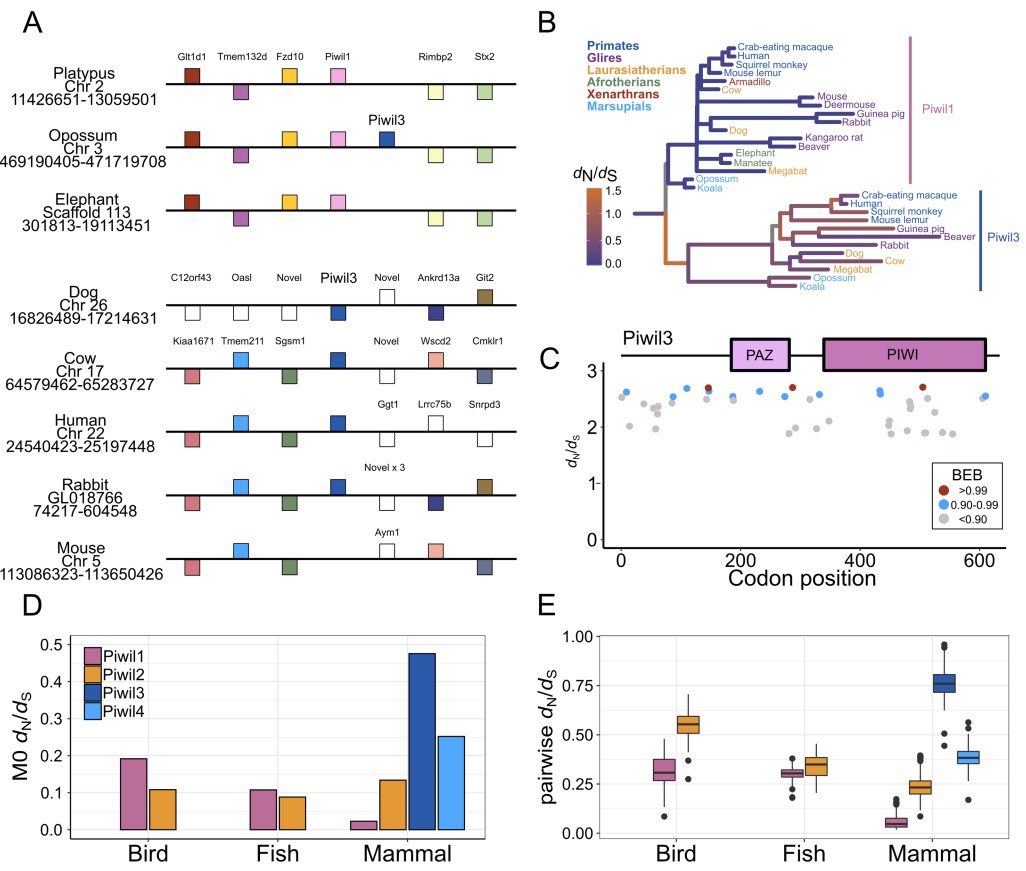

**Figure 3  Evolutionary properties of Piwil3.** (A) Synteny around the Piwil3 gene. White boxes represent genes that are not homologous to any other genes in the synteny block, and empty space between Sgsm1 and Aym1 in the mouse locus indicates a missing Piwil3. (B) Evolutionary rate measured as $d_N/d_S$ mapped to branches of a mammalian Piwil1 and Piwil3 phylogenetic tree calculated from a free-ratio model in Codeml. (C) Codon sites under selection estimated by Bayes Empirical Bayes (BEB) from Codeml model M2a. M2a sites were chosen over M8 because M2a is more conservative and sites under selection in M8 are also identified in M2a. (D) One-ratio $d_N/d_S$ estimates among Piwis in major lineages calculated using model M0 in Codeml. (E) Pairwise $d_N/d_S$ distances among Piwis and major lineages.

to saturation at synonymous sites, indicated Piwil3 is the fastest evolving Piwi among all lineages while Piwil1 in mammals is highly conserved.

## Historical Piwi expression

Our last goal was to identify any differences in Piwi expression among lineages, and consistent with previous results, Piwi expression was largely localized to the gonads, specifically testis, although Piwis were sporadically expressed at low levels among somatic tissues (Fig. 4A). *Aravin et al. (2008)* reported that Piwil4 is only expressed for a short period of time during testis development in mice, but Piwil4 has since been detected in adult undifferentiated spermatogonia (*Carrieri et al., 2017*; *Vasiliauskait et al., 2018*). Consistently, Piwil4 was detected in the adult testes of all species examined, but lowly detected in the mouse (Fig. 4B). In addition, we detected weak expression of Piwil3 in

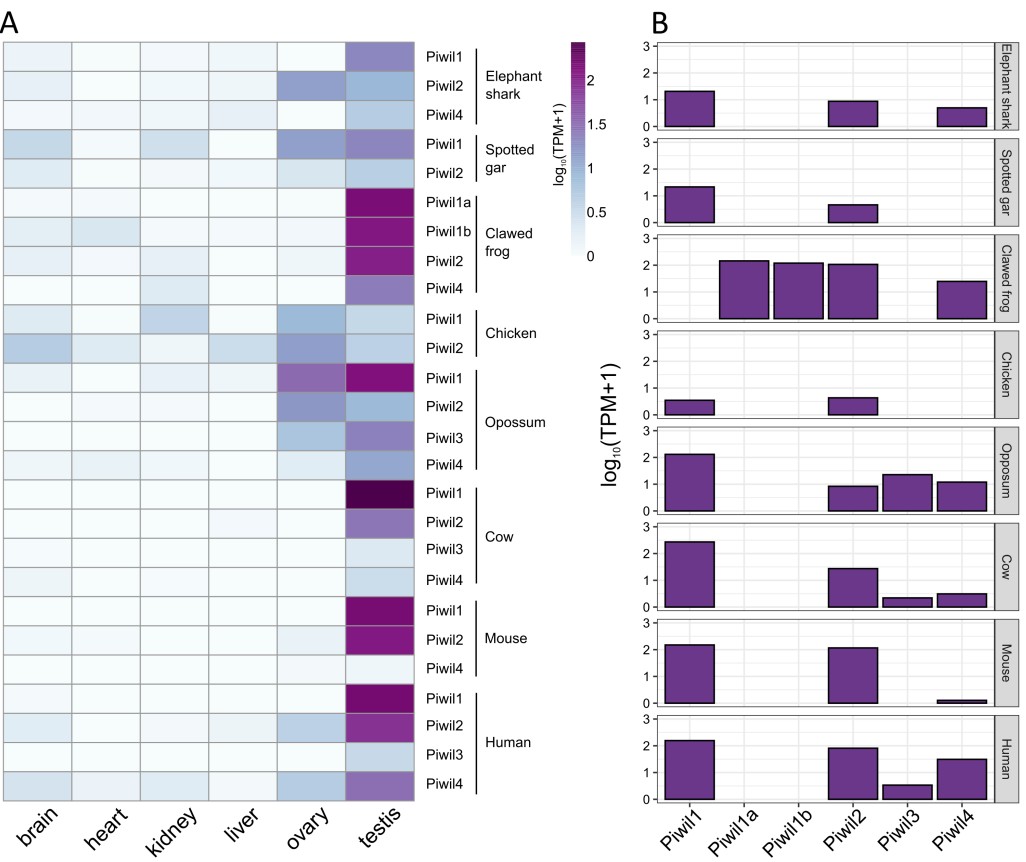

**Figure 4  Piwi expression characteristics.** (A) Expression of Piwi genes from both somatic and gonadal tissue from representatives of vertebrate lineages. Expression values were normalized by taking the $\log_{10}$ of transcripts per million (TPM) +1. (B) Bar plot of Piwi expression in just testes.

the testis of the opossum, human, and cattle. Both Piwil1 paralogs were expressed in the testis of the clawed frog (Fig. 4B). We were also able to measure Piwi expression from the brain, ovary, and testes, in the spiny toad which had an additional Piwil1 retrogene. The spiny toad expression pattern was unique as Piwil4 and Piwil1a-retro were the only Piwis expressed in the testis, but Piwil1a and Piwil1a-retro were expressed in the ovary (Fig. S4B). Piwil1b nor Piwil2 were expressed in any library (Fig. S4B).

## DISCUSSION

Understanding the relationships among gene family members is an active area of research in evolutionary biology and the availability of whole-genome sequences allows the opportunity to address unresolved questions. The Piwi gene family is a charismatic family of proteins that function as a genome defense mechanism against invasive elements that include TEs and viruses and function in a similar fashion as CRiSPR/Cas9. However, the paralog relationships and duplication timing in vertebrates has not been thoroughly resolved.

Consistent with previous work, Piwil2 was found to be monophyletic among deuterostomes (*Kerner et al., 2011*), but the Piwil1 group has experienced additional

duplications among vertebrates giving rise to the gnathostome Piwil4 and the therian Piwil3 paralogs (Fig. 1). Based on our phylogenetic and synteny analyses, we propose a model where Piwil4 originated from the duplication of Piwil1 early in vertebrate evolutionary history, at least as early as the ancestor of gnathostomes (Fig. 5). Unfortunately, the exact timing of Piwil4 origination is unresolved given that orthology could not be accurately inferred between cyclostomes and gnathostomes. Cyclostome genomes possess nucleotide biases and are the product of an additional cyclostome specific whole-genome triplication (*Nakatani et al., 2021*) and resolving orthology between gnathostome and cyclostome genes is often difficult (*Qiu et al., 2011*; *Kuraku, 2013*; *Smith et al., 2013*; *Campanini et al., 2015*; *Opazo et al., 2015*). However, given that 1- Piwil4 is secondarily lost, 2- all vertebrates have a Piwil1, and 3- there are only two Piwi paralogs in the lamprey genome, we propose that Piwil4 probably emerged in the gnathostome lineage, after the divergence between gnathostomes and cyclostomes. Piwil4 was then secondarily lost in the common ancestor of ray-finned fishes, retained in Sarcopterygii and its descendants, lobe-finned fishes, Sarcopterygii, and its descendants, but lost again in birds and monotremes (Fig. 5). Piwil3 originated in the common ancestor marsupial and placental mammals as a tandem duplication, but the retention and position of Piwil3 is variable among placental mammals (Fig. 3A).

Piwil3 was not identified in the elephant or armadillo genome. Under the most parsimonious scenario, Xenarthra and Afrotheria belong to the monophyletic group Atlantogenata (*Foley, Springer & Teeling, 2016*), and Piwil3 was lost in the common ancestor (Fig. S1). In addition, Piwil3 was also secondarily lost in some mouse-like rodents. Among Glires, rabbits, guinea pigs and beavers have retained Piwil3 (Fig. 3B), while the deer mouse, kangaroo rat, and house mouse lost Piwil3. The beaver is nested within the mouse-like rodent clade that includes the deer mouse, house mouse, and kangaroo rat (*Blanga-Kanfi et al., 2009*; *Fabre et al., 2012*), and encodes a Piwil3 while the remaining three species lack Piwil3 (Fig. 3B). This points to a complex scenario of gene retention (See Fig. S1) that would require further investigation to unveil.

We performed synteny analyses in an attempt to identify the duplication mechanism, be it whole genome, segmental, or tandem duplication. Piwil1 and Piwil2 were already present in the ancestor of deuterostomes so these paralogs are the product of a duplication early in animal evolution (*Kerner et al., 2011*), and we could not detect any co-duplicating gene families neighboring Piwil1 or Piwil4 (Fig. 2) that would point to segmental or whole genome duplication (*Catchen, Conery & Postlethwait, 2009*). In addition, Piwil3 is a direct neighbor of Piwil1 in the opossum genome, which likely reflects the ancestral state. Piwil3 likely migrated to a novel locus through non-homologous recombination in the ancestor of placental mammals, and synteny has not been conserved (Fig. 3A). From this evidence we conclude tandem duplication of Piwil1 followed by non-homologous recombination drove the expansion of the Piwi gene family.

The additional duplication events within the Piwil1 group likely led to novel functions among paralogs. PIWI proteins in mice bind piRNAs of different sizes and knockouts among paralogs yield discrete phenotypes (*Aravin et al., 2007*; *Aravin et al., 2008*, Kuramochi-Miyagawa et al. 2008). Unfortunately the exact function of all paralogs in all major

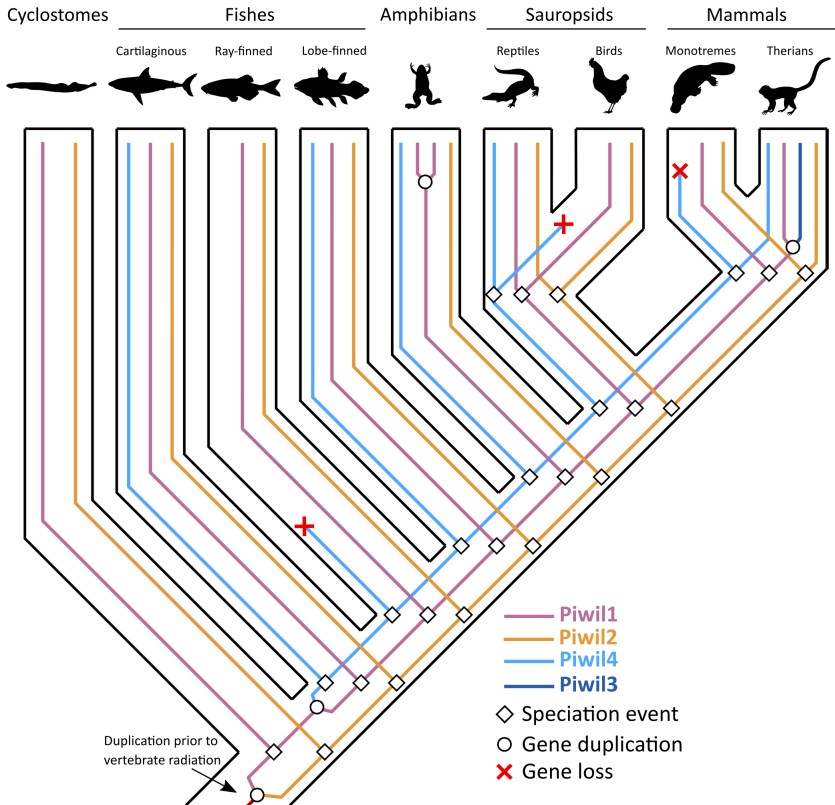

**Figure 5** **An evolutionary hypothesis describing the history of gene gain and loss among vertebrates.** The common ancestor of vertebrates had two Piwis (an ancestral Piwil1 and Piwil2). Piwil2 has not undergone any gain or loss during vertebrate evolution. Most likely, Piwil4 was derived from a tandem duplication of Piwil1 in the common ancestor of gnathostomes, but was independently lost in ray-finned fishes, birds, and monotreme mammals. Piwil3 was derived from a tandem duplication of Piwil1 in the common ancestor of therian mammals. Source credits: Lamprey/cyclostome, Gareth Monger, http://phylopic.org/image/f9313512-3ce4-4349-86bc-060d4faa013e/ (CC 3.0, http://creativecommons.org/licenses/by/3.0/); Lobe-finned fish, Maija Karala, http://phylopic.org/image/202c2ad3-48a7-471d-87ef-c6d8406640e8/ (CC NC 3.0, http://creativecommons.org/licenses/by-nc-sa/3.0/); Amphibian, Sarah Werning, http://phylopic.org/image/cd0f49a1-4adf-448e-859c-b703a73b9481/ (CC 3.0, http://creativecommons.org/licenses/by/3.0/) Monotreme, Sarah Werning, http://phylopic.org/image/cd0f49a1-4adf-448e-859c-b703a73b9481/ (CC 3.0, http://phylopic.org/image/b406c409-2735-4a3d-a7aa-8afe0b6e72dc/); Therian, Bogdan Bocianowski, http://phylopic.org/image/2d078b25-e6a0-4beb-a5d3-5d6f16be8ebf/) (CC 3.0, http://creativecommons.org/licenses/by/3.0/); Cartilaginous fish, Public Domain: http://phylopic.org/image/545d45f0-0dd1-4cfd-aad6-2b835223ea0d/; ray-finned fish, Public Domain: http://phylopic.org/image/6f4c653a-1da1-4e02-85ef-6344b2d8e02a/; Reptile, Public Domain: http://phylopic.org/image/dffda000-77cb-4251-b837-0cd2ab21ed5b/; Chicken, Public Domain: http://phylopic.org/image/aff847b0-ecbd-4d41-98ce-665921a6d96e/.

lineages is unknown, but expression initiation varies among lineages which points to slightly different functions. For example, Piwil1 expression begins in primordial germ cells at E7 in zebrafish (*Hsu et al., 2018*) and E6 in chicken (*Kim et al., 2012*), but Piwil1 expression begins during meiosis at P14 in mice (*Girard et al., 2006*). Results from our $d_N/d_S$ analyses suggest differences in selective pressure that could be related to these functional differences. Given the expression and $d_N/d_S$ differences, we can hypothesize

that the function of PIWIL1 removing mRNAs in later stages of meiosis could be unique to mammals. We also hypothesize there may be some functional redundancy between PIWIL2 and PIWIL4. Because PIWIL4 localizes to the nucleus and PIWIL2 localizes to the cytoplasm, it is thought that PIWIL2 is responsible for loading PIWIL4 with secondary piRNAs and PIWIL4 "marks" TE loci for subsequent methylation (*Carmell et al., 2007*; *Aravin et al., 2008*). Interestingly, there is some evidence that both PIWIL2 and PIWIL4 function to silence TE expression through methylation pathways (*Manakov et al., 2015*), although this claim is contested in Zoch et al. (2020). However, PIWIL2 does localize to the nucleus in zebrafish (*Houwing, Berezikov & Ketting, 2008*) and although untested, could potentially play a role in TE methylation (*Goll & Halpern, 2011*). Since both ray-finned fishes and birds lack Piwil4, it would be of interest to determine if the lack of Piwil4 has resulted in any measurable effect on TE methylation or test a methylation initiation function for PIWIL2 in these lineages.

Selection tests revealed evidence that Piwil3 is evolving in a fast and diversifying manner in all mammalian lineages (Fig. 3B; Fig. S8) and no Piwi in any other tested lineage is evolving in a diversifying manner (Table S3). The fast rate of Piwil3 was previously revealed in simian primates (*Wynant, Santos & Vanden Broeck, 2017*), but we demonstrated this phenomenon is not exclusive to primates. It is unclear why Piwil3 would exhibit an increased rate of evolution. Genes that tend to evolve in this fashion are often associated with immunity/defense against pathogens and reproduction/sexual selection (*Kosiol et al., 2008*; *Park et al., 2011*; *Vandewege et al. 2013*; *Grueber, Wallis & Jamieson, 2014*; *Vandewege, Sotero-Caio & Phillips, 2020*). Further, RNAi/antiviral genes have higher rates of evolution in insects relative to other genes (*Obbard et al., 2009*; *Wynant, Santos & Vanden Broeck, 2017*). Piwil1 and Piwil2, among other piRNA pathway genes, have high $d_N/d_S$ rates and appear to be evolving under positive selection in teleost fishes that have wide TE diversity (*Yi et al., 2014*). Since Piwil3 is possibly part of a genome immune system, it could be plausible Piwil3 adapts to lineage specific genome stressors. However, Piwil3 is not present in all mammalian lineages and mutant knockout experiments have not been conducted to understand its precise function, so there is currently no good explanation as to why Piwil3 is evolving so fast.

In general, our expression results were consistent with the literature. Piwis exhibited the highest expression in germline tissues (Fig. 4A) and Piwil1 and Piwil2 were the most expressed in adult testis (Fig. 4B). We also detected the expression of Piwil4 in adult testis among all gnathostomes that encode a Piwil4, albeit expression was lowest in the mouse (Fig. 4B). In mice, Piwil4 expression coincides with gonocyte methylation erasure and re-establishment (*Aravin et al., 2008*; *Carmell, 2008*; *Molaro et al., 2014*) where the function is linked to homeostatic and regenerative spermatogenesis (*Carrieri et al., 2017*; *Vasiliauskait et al., 2018*). There were some inconsistencies regarding ovarian Piwi expression. Piwi expression has been previously detected in the clawed frog (*Wilczynska et al., 2009*) zebrafish (*Houwing et al., 2007*; *Houwing, Berezikov & Ketting, 2008*), humans and cattle (*Roovers et al. 2015*). Piwil3 has only been described in maturing oocytes and early embryos (*Tan et al., 2020*) but we found it expressed in testis (Fig. 4A) although it has not been described in testis. Nonetheless, all four Piwis were expressed in both opossum

gonads suggesting Piwil3 localization to gonads was already established in the common ancestor of therian mammals.

## CONCLUSIONS

We present an evolutionary study of the Piwi gene family in vertebrates that is generally composed of two to four paralogs with emphasis on resolving the phylogenetic relationships and duplication history. Our study included representative species from all of the major groups of vertebrates and deuterostome outgroups. Piwil2 has not undergone any additional duplications through vertebrate evolution, but Piwil4 and Piwil3 are derived from independent duplications of the vertebrate Piwil1 at different times. Piwil4 likely first appeared in the ancestor of gnathostomes, but was secondarily lost in ray-finned fishes and birds, and Piwil3 is therian specific. The most likely mechanism yielding these novel paralogs was tandem duplication followed by non-homologous recombination. Piwi genes are largely evolving at different rates among lineages and expression has largely been constrained to the gonads through all of vertebrate history.

## ACKNOWLEDGEMENTS

The color palette was inspired by Prince and the Revolution's Purple Rain (1984).

### Funding
Support for this work came from a US Dept. of Education HSI-STEM grant P031C110114-15. The funders had no role in study design, data collection and analysis, decision to publish, or preparation of the manuscript.

### Grant Disclosures
The following grant information was disclosed by the authors:
US Dept. of Education HSI-STEM: P031C110114-15.

### Competing Interests
David Ray is an Academic Editor for PeerJ.

### Author Contributions

- Javier Gutierrez performed the experiments, analyzed the data, prepared figures and/or tables, authored or reviewed drafts of the paper, and approved the final draft.
- Roy Platt, Juan C. Opazo, David A. Ray and Federico Hoffmann conceived and designed the experiments, authored or reviewed drafts of the paper, and approved the final draft.
- Michael Vandewege conceived and designed the experiments, prepared figures and/or tables, authored or reviewed drafts of the paper, and approved the final draft.

## Data Availability

The SRA libraries used in expression analyses (Table S2) are available at Genbank.

The code used in this study is available at GitHub: github.com/mike2vandy/PiwiGeneFamily.

## Supplemental Information

Supplemental information for this article can be found online at http://dx.doi.org/10.7717/peerj.12451#supplemental-information.

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
