# Peer review of "Evolutionary history of the vertebrate Piwi gene family"

_PeerJ, doi:10.7717/peerj.12451_

## Round 0.1 · original submission · Major Revisions

Dear Dr. Gutierrez and colleagues:

Thanks for submitting your manuscript to PeerJ. I have now received three independent reviews of your work, and as you will see, the reviewers raised some concerns about the research. Despite this, these reviewers are optimistic about your work and the potential impact it will have on research studying Piwi systems and the evolution of mobile genetic elements. Thus, I encourage you to revise your manuscript, accordingly, taking into account all of the concerns raised by all three reviewers.

While the concerns of the reviewers are relatively minor, this is a major revision to ensure that the original reviewers have a chance to evaluate your responses to their concerns.

There are many comments by the reviewers that ask for more information on specific issues; please address these. Please consider providing more clarity on the various taxonomic groups that are discussed. The reviewers seem to have expertise on these and related systems, so their suggestions will improve your manuscript. There are also suggestions for literature very pertinent to your work. Finally, the tests for positive selection should be conducted over evolutionary groups only if saturation can be avoided.

I look forward to seeing your revision, and thanks again for submitting your work to PeerJ.

Good luck with your revision,


Best,

-joe

Reviewer 1 ·

Basic reporting

The paper provides a detailed phylogenetic analysis of PIWI Argonaute proteins across vertebrates. The analyses reveal interesting evolutionary turnover of PIWIs across vertebrates, significantly expanding on previous work.

In general, the authors had good language and a sufficiently detailed introduction, background and discussion. However, I suggest that authors elaborate/clarify evolutionary jargon to make the work more easy to read and accessible to all scientists. For example, it would be useful to define different clades (e.g. gnathostomes vs cyclostomes) at least once in the text or to use a species tree as a supplemental figure to label different clades.

I would also recommend the following changes that could improve ease of understanding the data:
1. Figure 1 is a bit overwhelming and difficult to read. The use of too many colors, especially since some colors overlap (for example, piwil1 and cartilaginous fish) make the figure more tedious to understand. I would suggest simplifying the figure by using only the color scheme for piwis or the major vertebrate clade. My preference as a reader was for the former (piwis colored). Alternatively, the authors could use box brackets next to the tree to indicate PIWIs without using colors. If using color for PIWIs, I would recommend a uniform color scheme across the paper. In addition, I would have appreciated an additional main or supplemental figure to summarize the presence or absence of a piwi in each vertebrate clade to complement Figure 1 (a simplified schematized version of Supplemental Table S1) similar to the last figure.

2. More descriptive figure legends would help the reader understand all aspects of the figure better. For example, in Figure 2, empty spaces for piwi that are meant to represent lack of evidence for presence or absence of the piwi. While this was described in the text, it would be useful to also include this in the legend.

Minor comments:
1. Simplify bootstrap representation in Figure 1 to just a % number and explain this in the legend as well.

2. The raw data provided was sufficient. My only recommendation would be to make labels in trees and alignments match that of the figures in the paper to allow for easy comparison.

Experimental design

The research covered in this manuscript are within the scope of the journal and addresses an open question in the field. The phylogenetic analyses span many vertebrate organisms revealing diversification of PIWIs. The methods section and supplemental data are largely sufficient, but some additional clarification and inclusion of some additional data would improve the presented work.

1. The phylogenetic trees constructed (Figure 1 and Supplementary Figure 1) use the entire sequence of PIWIs. However, N terminal residues are poorly conserved based on the alignment provided in the raw data for each tree. Poor alignment could lead artificial groups in the tree; therefore, I suggest using the most well aligned regions of the sequences to make a tree. The alignment submitted suggests these might be the PAZ and PIWI domains but not the N-terminal regions. A tree made only on these well aligned regions could be included as a supplementary figure.

2. The positive selection analyses presented in the paper span larger vertebrate clades. Since these species (for example, mammals) are very diverged, their dS tend towards saturation and can distort the omega calculation for these analyses. I would suggest also including a more closely related set of mammals (for example simian primates) in their analyses in Figure 3. Similar analyses have been previously done in Wynant et al., 2017 (https://doi.org/10.1038/s41598-017-08043-5), where they also suggest Piwil3 shows signatures of positive selection. However, with more available primate genomes now, a more comprehensive analysis (with more simian primates) would be informative. I would also suggest that authors reference Wynant et al., 2017 in their manuscript.

3. A nice addition to the current analyses might be to show the sequence divergence/ identity between PIWIs. A LOGOs plot spanning a fixed set of species (particularly in the PIWI and PAZ domains) or even a schematic showing differences in protein length with % identity labelled will be informative.

4. Similarly, it would be useful to actually see the positively selected residues (with BEB>0.9) across a group of species for Figure 3C. For example, a figure or table that shows each positively selected residue and how they have changed across a group of species to see the range of residues that can be found at each position.

5. From the descriptions in the text and the methods, it is not clear if the authors did a thorough analyses to identify all pseudogenes of PIWIs (other than for some species in 2B). Regardless, it would be informative to add details about if and how pseudogenes were identified in the sequence acquisition section in methods similar to their explanation in lines 204-208.

Validity of the findings

1. Since PIWIs appear to be expressed in ovaries (even if at lower levels than in testes), I suggest including a paragraph in the discussion on PIWI functions in the ovaries.

2. In the discussion section, a possibility that is not addressed is one where PIWIs can have redundant functions. Despite the divergence in sequence, I wonder if it is possible that in cases where a PIWI is pseudogenized or missing, it is because an alternate PIWI is functionally redundant even if this is species-specific. I suggest that the authors raise this possibility in the discussion and cite existing literature (if any) to support/oppose this hypothesis.

Additional comments

I recommend that the authors ask colleagues in both the evolution and PIWI field to read and comment on their manuscript to improve its readability for a broader audience. I would also like to commend the authors on their analyses during the difficult circumstances of the COVID-19 pandemic.

Reviewer 2 ·

Basic reporting

See general comments

Experimental design

See general comments

Validity of the findings

See general comments

Additional comments

Guiterez et al employ phylogenetics, synteny, evolution rate and expression to explore the evolution of vertebrate Piwi genes. I am in agreement with their findings and support publication. I appreciate the authors expertise does not primarily reside in the piRNA pathway and as such some of the statements regarding PIWI function require revision to incorporate more recent findings.

Major concerns
I think the evolution of Piwil2 should be incorporated into Figure 5.

PIWI Literature
1. The following statement requires revision: ‘Piwis are primarily expressed in gonads and protect the germline against the mobilization and propagation of transposable elements (TEs) through transcript cleavage and genome methylation.’. Not all animals utilise DNA methylation to transcriptionally silence TEs. Maybe one could utilise the term ‘transcriptional gene silencing’ and explain the role of DNA methylation in mammals.

2. The following statement requires revision: ‘PIWI proteins are generally restricted to the germline, bind their own class of small RNAs, PIWI interacting RNAs (piRNAs), and regulate TEs and viruses through transcript cleavage and genome methylation (Girard et al 2006; Lau et al. 2006; Brennecke et al. 2007).’. Same criticism as point 1.

3. The following statement requires revision: ‘PIWIs will then cleave the newly bound transcript and repeat the cycle. These piRNAs can also be used to guide PIWI proteins to TE loci in the genome and initiate methylation (Aravin et al. 2008; Kuramochi-Miyagawa et al. 2008; Rozhkov et al. 2013).’ The above sentence is not accurate. PIWIL4 is not an endonuclease (PMID: 22020280) and does not engage in ping-pong cycle. Phase piRNA biogenesis should be incorporated into the sentence (reviewed in PMID: 30446728 and original refs found within).

4. The following statement requires revision: ‘From these experiments, we understand the expression of Piwis varies temporally during embryonic and testis development.’. The following sentence would be more accurate: ‘From these experiments, we understand the expression of Piwis varies during germ cell development and spermatogenesis.’.

5. The following statement requires revision: ‘Piwil4 (Miwi2) is expressed for a short period of time between E14.5 and post-natal day 3 (P3), complements PIWIL2 in the ping-pong cycle, and is linked to the de novo establishment of methylation marks among TE loci in primordial germ cells (Carmell et al. 2007; Aravin et al. 2008; Molaro et al. 2014).’ As point 3. De novo methylation occurs in foetal gonocytes not PGCs. PIWIL4 is also expressed in a population of undifferentiated spermatogonia in the adult mouse testis (PMID: 28461596 and PMID: 29728652). I would add the following reference (PMID: 32674113) to above sentence which shows a detailed timeline moue PIWIL4 expression during germ cell development.

6. The following statement requires revision: ‘Piwil4 was detected in the testes of all species examined except the mouse (Fig. 4B). Aravin et al. (2008) reported that Piwil4 is only expressed for a short period of time during testis development in mice, but that does not appear to be universal among vertebrates.’. As point 5.

7. The following statement requires revision: ‘Because PIWIL4 localizes to the nucleus and PIWIL2 localizes to the cytoplasm, it was thought that PIWIL2 was responsible for loading PIWIL4 with piRNAs and PIWIL4 “marks” TE loci for subsequent methylation (Carmell et al. 2007; Aravin et al. 2008). However, it was later shown that both PIWIL2 and PIWIL4 function independently in methylation pathways and PIWIL2 initiates the methylation of more TE loci than PIWIL4 (Manakov et al. 2015).’. The second sentence should be revised in the light of a more recent paper (PMID: 28530707). I suggest eliminating the second sentence, how can a cytoplasmic protein function in the nucleus. The discrepancy arises between the purity of germ cells isolated for DNA methylation analysis.

8. The following statement requires revision: ‘For example, in mice, the expression of Piwil4 coincides with primordial germ cell methylation erasure and re-establishment (Aravin et al. 2008; Carmell 2008; Molaro et al. 2014), which led authors to suggest Piwil4 is essential to the de novo re-establishment of TE methylation in primordial germ cells.’. PIWIL4 is only expressed during the period of de novo methylation during germ cell development (see & also cite PMID: 32674113), this occurs in gonocytes not PGCs. The other important evidence is that PIWIL4 is a nuclear protein.

9. The following statement requires revision: ‘In contrast, our expression analyses suggest Piwil4 is also expressed in mature, adult testes of all other gnathostomes (Fig. 4B). It is currently unclear what the function of PIWIL4 exactly is in adult testes since this protein is no longer synthesized three days after birth in mice.’. PIWIL4 is expressed in a tiny population of undifferentiated spermatongonia (PMID: 28461596 and PMID: 29728652) but its function is dispensable for homeostatic and regenerative spermatogenesis (PMID: 29728652). Also PIWIL4 is expressed in the lung (PMID: 28920925).

10. In my opinion the following statement is not supported: ‘Therefore, the expression of Piwil4 in mouse primordial germ cells during methylation re-establishment could be coincidental, lineage specific, or a temporal function.’. The literature suggests that the primary function of PIWIL4 is during germ cell development and to instruct piRNA-directed DNA methylation.

Reviewer 3 ·

Basic reporting

The manuscript is well written, and enjoyable to read. The central message(s) is (are) simple and clear, and the manuscript makes a useful addition to the literature. The structure seems reasonable, the figures are excellent (I particularly like figure 5), and the raw data are all available as supporting material.

The one major issue I have with the writing is the lack of reference to the evolutionary literature; a casual reader is given no context by the present study. There are many papers on the evolution of RNAi pathways (including the piwi genes, e.g. https://doi.org/10.1093/gbe/evw018, https://doi.org/10.3389/fcimb.2021.653695, https://doi.org/10.1371/journal.pntd.0007919), and none of this literature is mentioned. The present manuscript is certainly novel - it is not duplicating previous work, which is primarily focussed on invertebrate taxa and plants - but it is odd that this large body of literature is not even mentioned.

This is even more striking given the analysis of evolutionary rates, as that has been a major focus of research on the evolution of RNAi, including in piwi-family genes (e.g. https://doi.org/10.1534/genetics.117.300567, and many references therein). For example, lines 314-316 note that rapid evolution is often associated with defence and reproduction, but this has been discussed in the context of piwis on many occasions (starting with https://doi.org/10.1098/rstb.2008.0168).

I understand that perhaps the authors wanted to keep things brief by not mentioning the invertebrate literature. However, this omission causes the introduction to become misleading in places. For example, the abstract and introduction state more than once that piwi expression is germline- and testis-biased. But this is an idiosyncrasy of vertebrates (and Drosophila!). It is not true of most animal phyla (e.g. https://doi.org/10.1038/s41559-017-0403-4, https://doi.org/10.1371/journal.pgen.1007533). Similarly, where the authors say that the “majority of animal genomes encode at least two copies of the Piwi family, Piwil1 (-iwi) and Piwil2 (-ili),”, this is not technically correct. Most animals (however you count) are probably nematodes, and most of the others are beetles. Most nematodes do not have these homologs, and the manuscript is very unclear on the homology with insect piwis. I don’t think it would even be true of most animal phyla? Are these two vertebrate piwi clades orthologous with Piwi/Aub and Ago3 (of insects)? I think they are not, but no literature is cited (e.g. https://doi.org/10.3389/fcimb.2021.653695, and references therein). If the intention was to ignore the invertebrate literature (which I think would be a mistake), it is very odd that the authors mention an antiviral role for piwis, as this is only known for a recent duplication in Aedes mosquitoes (https://doi.org/10.1371/journal.pntd.0007919)

Experimental design

With two exceptions (see ‘3. Validity’), the approaches seem reasonable, although the methods are a little light on explanation in places (did the Bayesian analysis not have any rate heterogeneity? Will all readers know what GTR+F+R7 is? From memory, GTR with estimated frequencies and an 7-category discrete rate variation?).

- How were the invertebrate sequences chosen? (I think it would be really useful to identify homology with the non-bilaterian piwis and ecdysozoan piwis.)

- I think (e.g.) line 125 “reverse translated” is the wrong terminology, since the degeneracy of the genetic code means that there is not a 1:1 reverse translation.

- I would also suggest PAML model 8a (max category fixed at 1) as the null for model 8, rather than model 7 (one fewer categories), although this is a minor issue

- The actual method of inference for the time and location of duplications in the tree is not made explicit. It is essentially manually-inferred / informal parsimony, which is fine in this simple case, but more sophisticated methods of gene-tree species tree resolution are available (e.g. https://doi.org/10.1093/molbev/mst100, and the literature that derives from it)

Validity of the findings

There are a couple of things that I think need attention.

The first, and most serious, is the Nei-Gojobori pairwise analysis. First, at this level of divergence such a simple counting approach is likely to be saturated. Was any saturation seen? If so, how was it dealt with? Second, the pairwise distances were then analysed with ANOVA. This is not a legitimate use of ANOVA, as it would assume independence of observations and observations are not independent because regions of the tree will appear in the analysis on multiple occasions. The appropriate way to analyse these data would be to extend the PAML analysis, using a clade analysis and distinguishing between models using (e.g.) AIC or likelihood ratio tests. See, for example, https://doi.org/10.1093/gbe/evw018.

The second thing, which needs checking but is unlikely to be a serious problem, is the extent to which the underlying data have been ‘cleaned’, or at least manually checked to accuracy. For example, while skimming the supporting material I noticed:

- The PIWIL3_Opossum has a long repetitive 5' region “GPAP ….”, which does not have a homologous repeat in other taxa, and this cannot be correctly aligned

- The mam piwi4 codon.aln file has several internal stops in XM_023739852.1_PIWIL4_Manatee, which cannot be correct (unless this is thought to be a pseudogene? But then it should probably not be included in analyses of the rate of evolution)

- At first glance, the mam.codon.fas for the site analysis has Piwil3 from a Megabat that is totally unaligned with the other sequences. I think there must be an error here?

- For the branch site analysis file mam.codon.aln the armadillo Pwil1 is clearly mis-spliced in that it has a retained intron.

On a more minor note, in lines 320-322, I think the authors have misunderstood the issue with recombination. Recombination inflates rates when there is recombination among the sequences used to build the tree, because by forcing relationships to a single tree (when there would in reality be one more tree topologies than recombination breakpoints) would generate homoplasy. Recombination per se (or nearby) is not a problem, and would not inflate rates unless it actually affected these sequences in a way that meant their relationship cannot legitimately be captured by a tree

Additional comments

It would be really useful to make clear the homology with piwis in invertebrates, and to note which piwis are (or are not) likely to be involved in a ping-pong process.

L84 – I would mention retrogenes in here too, since one is later found.

L207 – Was the putative pseudogene expressed?

---

## Round 0.2 · accepted · Accept

Dear Dr. Gutierrez and colleagues:

Thanks for revising your manuscript based on the concerns raised by the reviewers. I now believe that your manuscript is suitable for publication. Congratulations! I look forward to seeing this work in print, and I anticipate it being an important resource for groups studying Piwi systems and the evolution of mobile genetic elements. Thanks again for choosing PeerJ to publish such important work.

Best,

-joe